# Influence of Probiotic Strains *Bifidobacterium*, *Lactobacillus*, and *Enterococcus* on the Health Status and Weight Gain of Calves, and the Utilization of Nitrogenous Compounds

**DOI:** 10.3390/antibiotics11091273

**Published:** 2022-09-19

**Authors:** Luboš Zábranský, Anna Poborská, Branislav Gálik, Miloslav Šoch, Petr Brož, Martin Kantor, Naděžda Kernerová, Ivan Řezáč, Michal Rolinec, Ondrej Hanušovský, Ladislav Strnad, Nikola Havrdová

**Affiliations:** 1Department of Zootechnical Sciences, Faculty of Agriculture and Technology, University of South Bohemia in České Budějovice, Studentská 1668, 370 05 České Budějovice, Czech Republic; 2Faculty of Agrobiology and Food Resources, Institute of Nutrition and Genomics, Slovak University of Agriculture in Nitra, Trieda A. Hlinku 2, 949 76 Nitra, Slovakia

**Keywords:** blood parameters, dairy calves, diarrhea, feed additives, probiotics

## Abstract

The aim of this study was to monitor the effect of *Bifidobacterium bifidum* (BB) and the combination of *Lactobacillus sporogenes*, *Enterococcus faecium*, and *Bifidobacterium bifidum* (LEB) on the health status and weight gain of calves, and the utilisation of nitrogenous substances. The experiment was performed in the period from April 2020 to September 2020. A total of 90 Holstein heifers, which were one to 56 days old, were used as experimental animals. Differences in live weight gain were significant if we compared the LEB vs. BB group and the LEB vs. C, the control group (86.23 ± 5.49 kg vs. 84.72 ± 6.22 kg, *p* < 0.05; 86.23 ± 5.49 kg vs. 82.86 ± 5.35 kg, *p* < 0.01). Considering the live weight gain, group BB was heavier than group C only (84.72 ± 6.22 kg vs. 82.86 ± 5.35 kg, *p* < 0.05). An effect on reducing the incidence and duration of diarrheal diseases was not demonstrated in this study (*p* = 0.1957). The administration of feed additives had no statistically significant effect on the amount of N excreted in the feces. The values of hematological and biochemical parameters were unaffected except for the first sampling of urea. Other blood parameters were not affected by the addition of probiotic feed additives. The bacterial populations in the feces 5 days and 56 days after birth were not affected by the inclusion of feed additives.

## 1. Introduction

Diarrheal diseases are the most common and the most serious health problem of calves in the early postnatal period [1,2] affecting almost 19% of the animal population [3]. Katsoulos et al. [4] add that diarrhea is the most common cause of mortality in newborn calves. The most common pathogens causing diarrhea in calves are *Rotavirus* spp., *Coronavirus* spp., *Cryptosporidium parvum*, and *Escherichia coli*. In the case of such diarrhea, antibacterial, antiprotozoal, and immunomodulatory agents may be used for treatment. Otherwise, alternative treatments such as prebiotics and probiotics, synbiotics, or herbal mixtures and extracts can be used. Dietary supplements with vitamins and minerals also have supportive effects [2]. 

The definition of probiotics is as follows: “live microorganisms which when administered in adequate amounts confer a health benefit on the host” [5]. In the last two decades, the probiotic concept has also been applied to animal feed [6]. The ban on antibiotics as growth promoters is a challenge for animal nutrition and increases the need to find alternative methods. The use of probiotics and prebiotics in young ruminants is expanding as farmers look to use natural alternatives to antibiotics to help improve calf health and promote growth. [7]. Viable substitutes should be sought to help improve animal immunity [8]. According to Ohashi et al. [9], probiotics are now considered to be a potential alternative to antibiotics. Probiotics have the ability to maintain balance and activity of the intestinal microbiota and are considered to be beneficial to the host animal. They improve intestinal health stimulating the development of beneficial microflora, enhancing resistance to pathogenic bacteria, increasing the capacity of the digestive tract, lowering pH, and improving mucosal immunity. In adult ruminants, they have a positive effect on fiber and cellulose digestion [8,10]. The adhesion of probiotic bacteria to cellular receptors of enterocytes also initiates signaling events that lead to cytokine synthesis. The production of butyric acid by some beneficial bacteria further affects the conversion of enterocytes and neutralizes the activity of some carcinogenic substances such as nitrosamines. These are produced by the metabolic activity of commensal bacteria in organisms consuming a high protein diet [11]. Probiotics are commonly used in cattle breeding as growth promoters [12]. *Bifidobacterium species* are considered to be one of the key genus in the intestinal tract of animals and humans. Its presence and quantity is associated with the good health of the host. Representative species include *Bifodobacterium longum*, *Bifodobacterium breve*, *Bifidobacterium bifidum*, and *Bifidobacterium infantis* [13]. In ruminants, bacteria of the genera *Enterococcus* and *Lactobacillus* may help to reduce acidosis [10].

Ruminant animals play an important role in sustainable agricultural systems [14] and they are also an important source of greenhouse gas emissions [15]. Reviews of CH_4_ mitigation strategies consistently discuss the possibility that lactic acid bacteria (LAB) could be used to modulate rumen microbial communities, thus providing a practical and effective on-farm approach to reducing CH_4_ emissions from ruminant livestock [16,17,18,19]. 

The aim of this study was to monitor the effect of *Bifidobacterium bifidum* (BB) and the combination of *Lactobacillus sporogenes*, *Enterococcus faecium*, and *Bifidobacterium bifidum* (LEB) on the health status, weight gain of calves, and the utilization of N substances in the body.

## 2. Materials and Methods

### 2.1. Ethical Approval

All study procedures were approved in accordance with the “Act on the protection of animals used for scientific purposes” of the Czech Republic. This act is in accordance with the EU Directive (No. 2010/63/EU) on the protection of animals used for scientific purposes and with the decision of the Ministry of Agriculture of the Czech Republic No. 22036/2019-MZE-18134.

### 2.2. Animals and Basic Feed Ration

The experiment was performed between April 2020 and September 2020 in a commercial dairy farm (N 49°55′; E 14°21′) in the Czech Republic. A total of 90 Holstein heifers were studied from birth to 56 days of age, and used as experimental animals. The feed ration was adjusted according to nutritional requirements. Each experimental period lasted for 56 days. Sampling always took place on days 5 and 56 after birth. Immediately after the birth, the calves were moved to outdoor individual boxes, where they were housed until the age of 56 days. The calves were fed colostrum a maximum of two hours after birth. They were fed colostrum from their own mother or frozen colostrum twice a day. The amount was three to four liters per feeding. Colostrum and subsequent milk replacers were administered to calves in plastic buckets with nipples which were placed in outdoor individual boxes at a height of 40 cm above the ground. Starting from the 5th day after calving, the calves were fed native milk twice a day at a rate of 4–5 liters per feeding with ad libitum access to drinking water, granulated calf starter mixture, and hay. The calf starter was presented to the calves from the 4th day after the birth. The calves are fed milk replacer from the 13th day after birth. 

Composition of milk replacer: dried whey protein, mixture of vegetable oils (palm and coconut), hydrolyzed wheat gluten, calcium carbonate, and garlic. Analytical constituents: crude oils and fats 18%, crude protein 23%, crude fiber 0.0%, crude ash 7.5%, calcium 0.9%, sodium 0.4%, phosphorus 0.7%. Nutritional supplements: vitamin A 25,000 International Unit (IU)/kg, vitamin D3 10,000 IU/kg, vitamin E 500 IU/kg, potassium iodide—0.25 mg/kg, manganese sulphate monohydrate—40 mg/kg, copper sulphate pentahydrate—10 mg/kg, sodium selenite—0.4 mg/kg, ferrous sulphate monohydrate—100 mg/kg, and zinc sulphate monohydrate—50 mg/kg. Antioxidants E321 BHT 150 mg/kg. Citric acid preservative—1000 mg/kg.

Composition of the calf starter: wheat 5%, barley 20.14%, oats 8%, corn 17%, wheat bran 9%, premix 0.2%, soybean meal without GMO 24.5%, dry alfalfa 10%, sugar 1.5%, vegetable oil 1.5%, limestone 1.45%, salt 0.48%, vit. A—145,000 IU/kg, vit. D3—2700 IU/kg, anhydrous calcium iodate—1.30 mg/kg, copper sulphate pentahydrate—25 mg/kg, manganese oxide—60 mg/kg, zinc oxide—85 mg/kg, sodium selenite—0.50 mg/kg and vit. E as alpha-tocopherol—70 mg/kg.

### 2.3. Treatment of Newborn Calves and Veterinary Care

The calves were provided with proper basic care throughout the study. A carer checked or, in case of need, ensured the viability of the individual. After ensuring basic vital functions, the umbilical stump of calves was disinfected. Pederipra Spray (chlortetracycline spray for superficial wounds) was used to treat the navel. Each calf, treated and taken care of, was transported to a clean, disinfected outdoor individual box with straw bedding. To check if calves receive a sufficient amount of quality colostrum, the blood samples were taken from the *jugular vein* between the 3rd and 5th day after birth. Subsequently, the blood was centrifuged (2000 RPM) and the total protein level was determined from the obtained blood plasma with a digital refractometer.

In order to reduce the risk of injury to animals during struggles for social status and to eliminate the risk of injury to humans by animals, cauterization of calves was performed by a veterinarian using a gas cauter in 3–4-week-old calves. Thus, it was done according to Act 246/1992 Coll., The Act on the Protection of Animals against Cruelty.

Two types of vaccines have been used in breeding. The first was Hiprabovis Balanc inj. sicc. to us. vet. It is a trivalent vaccine for the active immunization of cattle against bovine parainfluenza virus-3 (PI-3), bovine viral diarrhea virus (BVDV), and bovine respiratory syncytial virus (BRSV). Two-month-old calves were vaccinated by receiving a dose of 3 mL intramuscularly. The procedure was followed by re-vaccination after 3 weeks. The second vaccine was Trichoben. This medicine is used regularly and in a preventive manner in calf rearing. It helps to reduce the clinical signs of dermatophytosis caused by the dermatophyte *Trychophyton verrucosum*. It is used for prophylactic vaccination as well as for therapeutic use. Using this vaccine, the calves are also vaccinated when two-month-olds with a dose of 2 mL IM, with the following re-vaccination after 2 weeks. Both vaccines were administered only after the end of the experiment.

### 2.4. Experimental Design

The experiment included 90 Holstein heifers from one dairy herd, which were divided immediately after birth into three groups according to date of birth—30 of them were in the group receiving *Bifidobacterium bifidum* (BB), 30 of them were in the group receiving a combination of *Lactobacillus sporogenes*, *Enterococcus faecium*, and *Bifidobacteriunm bifidum* (LEB), and 30 of calves were in the control group (C). Experimental group BB received orally 3 g of *Bifidobacterium bifidum* (4.1 × 10^7^ Colony Forming Units/g) orally in colostrum and subsequently in the milk replacer from the first to the 21st day after calving. The LEB experimental group received 3 g of *Lactobacillus sporogenes* (4.1 × 10^7^ CFU/g) + 1 g of *Enterococcus faecalis* (4.1 × 10^7^ CFU/g) + 1 g of *Bifidobacterium bifidum* (4.1 × 10^7^ CFU/g) orally in colostrum and subsequently in the milk replacer from the first to the 21st day after calving. The experimental groups were given these feed additives once a day during the second feeding and were not treated with antibiotics during the experiment. The control group received a basic feed ration without feed additives and was treated with antibiotics if necessary. In the case of severe diarrheal diseases, calves from the control group were given, intravenously, the antibiotic Borgal from the manufacturer Virbac SA (Carros, France). These antibiotics were administered by a veterinarian. Antibiotics were selected after culture for sensitivity. The result of determining the sensitivity of the tested bacteria to antimicrobial preparations is an antibiogram, which enables the selection of the most suitable antibiotic for treatment.

All calves were weighed within two hours after birth and on day 56 after birth. The classical method according to Larson et al. (1977) [20] was used to evaluate and detect diarrheal diseases. Observations of feces and health status were evaluated twice a day together with measurement of rectal temperature at the time of feeding. Respiratory status was assessed according to the types of symptoms (normal, runny nose, shortness of breath and cough–wet or dry). Respiratory disease was rated as occasional, intermittent, or persistent. The carers monitored the state of the hair and eyes (dullness and brightness) and signs of dehydration (sunken eyes, inelastic skin, and exhaustion).

### 2.5. Feed Intake Measurement

The intake of colostrum and later milk replacer was controlled by eSense Flex ear sensors from the manufacturer Allflex (Cape Town, South Africa). These sensors are connected to the Milk Taxi via Bluetooth technology. Each calf is weighed after birth and the consumption of milk replacer per feeding is set according to its live weight. Each calf receives the exact dose according to the set growth curve.

### 2.6. Weighing Calves

The calves were weighed during transport from the maternity ward to outdoor individual boxes no later than 2 hours after birth and then on the 56th day after birth during transport to group boxes in the calf housing. A two-wheeled cart with built-in strain gauges with an accuracy of 0.01 kg was used for weighing and transport. The cart was also equipped with a fixation barrier and was used for the cauterization of the calves. A Pedoza Rhino cart with tensometric scales from the manufacturer ForstAgro (Pelhrimov, Czech Republic) was used for weighing the calves.

### 2.7. Fecal Sampling 

Fecal samples were taken from each calf on the 5th and 56th day after birth. Feces for the analysis of total nitrogen were collected in plastic containers with screw caps. Feces for microbiological analysis were collected from the rectum of calves with pre-prepared sterilized cotton swabs at a depth of 5 cm and placed in a closed sterilized tube. The tubes were sealed immediately after swabbing to prevent environmental contamination. All samples were immediately put in a refrigerator, stored at a constant temperature of 4 °C, and transported to a certified laboratory. Fecal samples for the determination of total nitrogen were freeze-dried for 72 h using a Heto PowerDry LL3000 freeze dryer, manufacturer Thermo Fisher Scientific (Waltham, MA, USA) and then crushed in a mortar. The N content of the feces was determined using the Kjeldahl method of AOAC (2005) No. 981.13 and the CP content was calculated as N × 6.25.

### 2.8. Blood Sampling

Blood samples were taken from the *jugular vein* every morning at 6:00 on the 4th and 21st day after birth. There was 5 mL of blood collected in tubes containing anticoagulant (mixture of sodium EDTA and sodium fluoride) and 5 mL was collected in an anticoagulant tube NTS. Blood samples were placed in a cooling box and processed in a laboratory within 2 h, where a biochemical analysis was performed using an Ellipse biochemical analyzer, manufacturer Dialab (Wiener Neudorf, Austria). The blood count was evaluated using an Exigo LABtechnik hematology analyzer, manufacturer LABtechnik (Brno, Czechia). The monitored parameters were: Hemoglobin, hematocrit, erythrocytes, leukocytes, glycemia, urea, alkaline phosphatase, gamma-glutamyl transferase, total protein, cholesterol, zinc, copper, phosphorus, calcium, and magnesium.

### 2.9. Statistical Analysis

Two statistical methods were determined to evaluate the results. The data were analyzed using a General Linear Model ANOVA (four ways with the interactions) of the statistical package STATISTICS 10 (Analytical Software, Tallahassee, FL, USA). Factors were evaluated of the treatment group (1—BB *Bifidobacterium bifidum*, *n* = 30, 2—LAB *Lactobacillus sporogenes*, *Enterococcus faecium* and *Bifidobacterium bifidum*
*n* = 30; and 3—C control, *n* = 30); Normality of data distribution was evaluated by Wilk–Shapiro/Rankin Plot procedure. All data conformed to a normal distribution. Significant differences between the groups were tested by comparisons of mean ranks. Values are expressed as means ± standard deviation and differences were considered significant at *p* < 0.05. Statistical analysis was performed using the general linear model procedures of SAS version 9.4 (SAS Inst. Inc., Cary, NC, USA) for a replicated latin square design using the model:Yijk = μ + Ti + Pj + Ck + εijk
where Yijk refers to the dependent variable; μ, overall mean; Ti, BB and LEB effect (i = 1, 2, 3, and 4); Pj, period effect (j = 1, 2, 3 and 4); Ck, steer effect (k = 1, 2, 3, 4, 5, 6, 7, and 8); εijk, residual error. All differences among the experimental diets were compared by Duncan’s multiple range test. Polynomial contrasts were conducted to determine the linear and quadratic effects of BB and LEB levels. Differences among the treatments were considered to be significant at *p* < 0.05 and were considered as a trend towards significance at 0.05 < *p* < 0.10.

## 3. Results

The effects of probiotic feed additives on live weight and weight gain of the calves are shown in Table 1. Calves from the LEB group reached the highest live weight on the 56th day. The differences were significant as compared to the group BB and the group C (86.23 ± 5.49 kg vs. 84.72 ± 6.22 kg, *p* < 0.05; 86.23 ± 5.49 kg vs. 82.86 ± 5.35 kg, *p* < 0.01). Considering the live weight gain, group BB was heavier than group C only (84.72 ± 6.22 kg vs. 82.86 ± 5.35 kg, *p* < 0.05). The application of antibiotics had a significantly smaller effect on the average gain in live weight on the 56th day of age in the control group C. This result confirms that the preventive use of probiotic feed additives has a beneficial effect on the health of the host. There was no difference between the groups in the incidence or duration of diarrhoea diseases.

The bacteria found in feces are shown in Table 2. *Escherichia coli* was the most common in the samples, as it occurred in all calves in the 1st and 2nd sampling of all the groups. The amount of *Campylobacter jejuni* in the 3 groups increased across the course of the study. *Morganella morganii*, a normal part of the intestine, can also be found in the surrounding environment. Infections are usually induced endogenously. Their pathogenicity is low. In healthy individuals, they are capable of causing infections of the urinary system at most, but also pneumonia, meningitis, or sepsis in weakened individuals. This bacterium was found only in the first blood draws. *Citrobacter* spp. and *Klebsiella pneumoniae* were also present in increased amounts. Considering other found organisms, only a rare occurrence was recorded. It is interesting that despite the addition of selected species of bacteria into colostrum and milk replacer, not a single applied species appeared in the results of fecal analyses. There was no difference between the 3 groups in the bacterial populations found in the calf feces.

The addition of BB and LEB to the feed ration had only a limited positive effect on the use of nitrogenous substances in dairy calves (Table 3), and no potential to positively affect the environment by reducing the production of total nitrogen in feces was demonstrated. The increased level of urea in the blood on the 4th day after birth in the LEB group can be explained by the better feeding of this group of calves on the day of sampling. For other blood parameters, the effect of adding BB and LEB to the feed ration on these parameters has not been demonstrated as shown in the Table 4.

The values of hematological and biochemical parameters were unaffected except for the first sampling of urea. The increase in serum urea nitrogen during the neonatal period it is mainly dependent on protein intake from colostrum and were not affected by the addition of probiotic feed additives. Other blood parameters were in balance and corresponded to the reference values. 

## 4. Discussion

The inclusion of both the probiotic feed additives in the diet of dairy calves influenced weight gain only, whereas there was no statistically significant effect on reducing the incidence of diarrheal diseases. Bayatkouhsar et al. [22] reports that the results of the present experiment indicated that the feeding of probiotics improved final body weight, daily BW gain, and starter intake. Moreover, the positive effect of probiotic use on the weight gain of calves was reported by Soto et al. [23], Frizzo et al. [24], and Timmerman et al. [25]. In contrast, Simon et al. [6], Uyeno et al. [10], and Renaud et al. [26] claim that the improvements in weight gain and feed conversion are only isolated elements and the effect of probiotics on health improvement was inconclusive. According to them, studies on this topic have yet been insufficient. The results of this study suggest that the supplementation of feed ration with probiotics does not have improving effects on the incidence of diarrheal diseases in calves, as reported by He et al. [27]. The incidence of diarrheal diseases is a major problem in calves, as confirmed by studies by Cho et al. [2], Smulski et al. [3], and Katsoulos et al. [4]. Ülger et al. [12] and Soto et al. [28] argue that regular, preventive administration of probiotics can lead to the improvement of calf health.

According to Martín et al. [29] and Nagy et al. [30], *Escherichia coli* is still considered to be a major infectious factor causing neonatal diarrhea in calves. Luginbühl et al. [31] disagree with this statement. They stated that *Escherichia coli* was less important in the studied populations in comparison to cryptosporidiosis and rotavirus infection. Barrington et al. [32] and Yonis et al. [33] found out that calves that received colostrum directly from their mother (unlike hand-fed calves) suffer from a smaller risk of infection of *Escherichia coli*. 

*Campylobacter jejuni* was another increased indicator which occurred to a greater extent in all groups of the observed groups of calves. Klein et al. [34] found the occurrence of *Campylobacter jejuni* in most of the studied individuals regardless of the incidence of diarrheal diseases and have stated that calves act more as a reservoir and could, therefore, infect other animals or humans. Besser et al. [35] reported a high incidence of *Campylobacter jejuni* in calves up to 4 months. They believe that the occurrence is supported by carry-over between the calves themselves. Furthermore, Sato et al. [36] found that, in comparison to adults, the incidence of *Campylobacter jejuni* is significantly increased in calves. 

*Citrobacter* spp. and *Klebsiella pneumoniae* were also present in increased amounts. Windeyer et al. [37] stated that these bacteria cause problems in calf breeding, such as neonatal septicemia, causing serious diseases and death of calves. Fecteau et al. [38] reported that infections are caused by the fecal-oral route and often during the first days after birth. Godden et al. [39], Vogels et al. [40], and Komine et al. [41] stated that the carry-over of infection is facilitated by the failure of passive immunity in calves.

The studies by Knowles et al. [21], Mohri et al. [42], Greenwood et al. [43], and Abeni et al. [44] argue that the amount of urea varies and may be related to protein intake in feed. Ballou et al. [45] found that *E. coli* has no effect on urea levels in the body.

Increased digestibility of total nitrogen and its reduced excretion in feces due to the application of probiotic strains of *Bifidobacterium*, *Lactobacillus*, and *Enterococcus* in the feed ration has not been proven. No publications dealing with the association between probiotic strains and N metabolism were found. There are several studies that suggest different strains of probiotics can be used to reduce CH_4_ production in ruminant livestock. Reviews of CH_4_ mitigation strategies consistently refer to this possibility Hristov et al. [16], Jeyanathan et al. [17], Knapp et al. [18], and Varnava et al. [19]. However, research on the topic has been limited and convincing data from animal trials to support this concept are lacking.

Regarding the other determined blood parameters, the effect of BB and LEB supplementation of the feed ration was not demonstrated. The increase in serum urea nitrogen during the neonatal period may be due to the increase in protein intake available through consumption of colostrum. Similar relationships have been reported in neonatal dairy calves during the first 3 d postnatally (Hammon et al. [46]). Although neonatal calf serum urea nitrogen is related to protein intake Steinhoff-Wagner et al. [47] and Rauprich et al. [48], this relationship can be inconsistent. Given that calves can deaminate amino acids when the resulting carbon skeleton is needed for gluconeogenesis (Hammon et al. [49]), energetic status of calves interacts with protein intake to cause the variation observed in later serum urea nitrogen concentrations. 

## 5. Conclusions

The addition of BB and LEB to colostrum and milk replacer had a demonstrable effect on live weight gain in calves which were one up to 56 days old. The tested feed additives had no statistically significant effect to reduce the incidence and frequency of diarrheal diseases in calves up to 56 days old (*p* = 0.1957). In comparison with the experimental groups LEB and BB, the results show an increased frequency of diarrheal diseases in group C which was fed only the basic feed ration. With the preventive use of probiotic and symbiotic feed additives, there was no need for the application of antibiotics for diarrheal diseases of the calves, and there was an improvement in live weight gains compared to the control group. Health status tended to be improved in experimental groups. The supplementation with probiotic of feed additives had no statistically significant effect on the content of nitrogen excreted in the feces or on the values of hematological and biochemical parameters. Only the level of urea on the 4th day after birth was proven to be significant different *p* = 0.044. However, this fact does not confirm the hypothesis that probiotic feed additives can reduce the levels of nitrogen produced in the feces which enter the environment.Probiotic supplementation to colostrum and milk replacer can improve the health and performance of calves.

## Figures and Tables

**Table 1 antibiotics-11-01273-t001:** The influence of applied additives on the growth and health of calves.

Variables	*n*	Treatment Groups			*p*	Significance
		BB	LEB	C		
		$\bar{x}$ ± SD	$\bar{x}$ ± SD	$\bar{x}$ ± SD		
Birth BW (kg)	90	47.81 ± 5.21	47.63 ± 5.08	47.76 ± 4.82	0.4642	
BW on 56th day (kg)	90	84.72 ± 6.22	86.23 ± 5.49	82.86 ± 5.35	0.0012 **	2:3 **, 2:1 *, 3:1 *
ADG from birth to 56th day (g)	90	369.1 ± 64.0	386.6 ± 75.0	351.0 ± 85.6	0.0012 **	2:3 **, 2:1 *, 3:1 *
Duration of diarrhea (in days)	90	1.69 ± 3.38	1.56 ± 3.28	1.89 ± 3.49	0.1957	
Total number of diarrheas	90	0.21 ± 0.37	0.18 ± 0.33	0.24 ± 0.39	0.0725	

* *p* < 0.05; ** *p* < 0.01; SD = standard deviation; ADG = average daily gain; BW = body weight; *p* = significance; *n* = number of calves (BB—*Bifidobacterium bifidum*, *n* = 30, LEB—*Lactobacillus sporogenes*, *Enterococcus faecalis*, *Bifidobacterium bifidum*, *n* = 30; and C–control, *n* = 30).

**Table 2 antibiotics-11-01273-t002:** Bacteria found in calf faces on the 5th and 56th day after birth.

	Treatment Groups					
Bacteria	BB		LEB		C	
	on 5th Day	on 56th Day	on 5th Day	on 56th Day	on 5th Day	on 56th Day
*Campylobacter jejuni*	+	++	+	++	+	++
*Citrobacter amalonaticus*	0	+	0	0	0	+
*Citrobacter freundii*	+	+	+	0	++	+
*Citrobacter koseri*	+	0	0	0	0	+
*Enterobacter kobei*	0	0	0	0	0	+
*Escherichia coli*	+++	+++	+++	+++	+++	+++
*Escherichia fergusonii*	0	0	+	0	0	0
*Morganella morganii*	++	0	++	0	++	0
*Klebsiella pneumoniae*	++	+	++	+	++	+
*Proteus miriabilis*	+	0	0	0	+	+
*Providencia stuartii*	+	0	0	0	0	0
*Proteus vulgaris*	+	0	0	0	+	0

+++ score 2.300–3.000, ++ score 2.000–2.299, + score 1.700–1.999, (BB—*Bifidobacterium bifidum*, *n* = 30, LEB—*Lactobacillus sporogenes*, *Enterococcus faecalis*, *Bifidobacterium bifidum*, *n* = 30, and C—control, *n* = 30).

**Table 3 antibiotics-11-01273-t003:** Nitrogen balance in feces and blood samples.

Attributes	*n*	Treatment Groups			*p*	Significance
		BB	LEB	C		
Nitrogen excretion in feces on 4th day (g/day)	90	18.55	18.79	18.63	0.089	NS
Nitrogen excretion in feces on 21st day (g/day)	90	23.21	23.88	24.12	0.059	NS
Urea concentration in blood on 4th day (mmol/L)	90	3.96	4.31	3.43	0.044	2:3 *
Range of urea in blood 21st day (mmol/L)	90	2.97	3.27	3.12	0.092	NS

*n* = number (BB—*Bifidobacterium bifidum*, *n* = 30, LEB—*Lactobacillus sporogenes*, *Enterococcus faecalis*, *Bifidobacterium bifidum*, *n* = 30; and C—control, *n* = 30). Means bearing different superscripts in a row differ significantly (* *p* < 0.05); NS, non-significant, *p* > 0.05.

**Table 4 antibiotics-11-01273-t004:** Average blood values of calves 4th and 21st day after birth.

Treatment Groups							
	Reference Value Units	BB 4.	LEB 4.	C 4.	BB 21.	LEB 21.	C 21.
Hemoglobin	g L^−1^ (94.6–130.8)	107.35	107.58	107.69	106.42	106.9	106.19
Hematocrit	L L^−1^ (0.27–0.37)	0.22	0.25	0.24	0.23	0.27	0.25
Erythrocytes	T L^−1^ (5.61–7.75)	5.3	5.21	5.01	5.36	5.65	5.48
Leukocytes	L L^−1^ (5.3–12.7)	6.66	6.29	6.09	8.35	8.18	8.14
Glycemia	mmol L^−1^ (3.4–6.1)	6.85	7.22	7.38	6.41	6.53	6.54
Urea	mmol L^−1^ (5.2–12.7)	3.96	4.31	3.43	2.97	3.27	3.12
Alkaline phosphatase	μkat L^−1^ (1.7–3.9)	5.78	5.31	5.69	4.39	4.17	4.19
Gamma-glutamyl transferase	μkat L^−1^ (0.67–8.29)	15.06	14.9	14.67	0.69	0.71	0.74
Total protein	g L^−1^ (51.9–67.3)	66.41	67.04	66.43	66.17	66.23	66.06
Cholesterol	mmol L^−1^ (2.6–4.6)	1.92	2.16	2.15	2.97	2.24	2.43
Zinc	mg L^−1^ (5.97–18.86)	28.78	29.02	29.08	24.5	24.75	23.9
Copper	mg L^−1^ (5.97–18.86)	13.93	13,72	13.55	14.98	15.08	15.05
Phosphorus	mmol L^−1^ (2.39–2.79)	3.00	3.2	3.2	3.33	3.49	3.39
Calcium	mmol L^−1^ (2.48–3.0)	3.16	3.10	3.14	2.94	2.8	2.91
Magnesium	mmol L^−1^ (0.72–0.94)	0.8	0.8	0.8	0.86	0.85	0.84

(BB—*Bifidobacterium bifidum*, LEB—*Lactobacillus sporogenes*, *Enterococcus faecalis*, *Bifidobacterium bifidum*, C—control, Reference value units—Knowles et al., 2000 [21]).

## Data Availability

All data are available in this study.

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
