# Peer review of "Influence of Probiotic Strains Bifidobacterium, Lactobacillus, and Enterococcus on the Health Status and Weight Gain of Calves, and the Utilization of Nitrogenous Compounds"

_antibiotics, 2022, doi:10.3390/antibiotics11091273_

Round 1
Reviewer 1 Report
General Comments
I found the manuscript to be of scientific interest. It describes a study in which an examination of the effect of different probiotic strains administered to calves.
The paper is relatively short and covers the results obtained reasonably well.
The title of the paper should be altered. I suggest the title should be "Influence of Probiotic Strains Bifidobacterium, Lactobacillus, 2 and Enterococcus on the Health 3 Status and Weight Gain of Calves, and the Utilization of Nitrogenous Compounds"
The statistical analyses performed are appropriate for the data sets.
The results were brief but clearly stated. I recommend more of the data gathered be presented in the Results section (see below).
The discussion covered much of the findings. I believe it needs to be re-written. I did not like the discussion about the use of antibiotics or the diarrhoea cases, which I will comment on in my specific comments. I suggest the Discussion and the Conclusions be re-written and much more attention is paid to the use of English.
Specific Comments
The title of the paper should be altered as suggested above. There is nothing in the paper that suggests antibiotic use is reduced. The study design required the treatment groups not to receive antibiotics. In saying that, I did not like this aspect of the study (as the control group were administered antibiotics as required). The authors must comment on this aspect of the study and provide some commentary on the severity of disease and the possible impact of disease and antibiotic use on the results.
Line 15 - should read "... health status and weight gain of calves, and the utilisation of nitrogenous substances."
Line 16 - should read " ... and the LEB vs C, the control group ... "
Line 20 - should read "... group BB was heavier than group C only .."
Line 22 - should read "... was not demonstrated in this study (P = 0.0725)."
Line 23 - The sentence starting on this line is ambiguous. Do the authors mean to say there was no difference in haematological and biochemical parameters measured in the 3 groups except for a higher urea concentration in the blood of the two treatment groups compared to the control group? It is difficult to interpret this as no results are shown from the haematology and biochemistry work other than 'urea'. If that is the case then the sentence could be written as above. The P value does not need to be provided here.
I strongly recommend the inclusion of the core haematology and biochemistry data in a simple results table in the Results section (and a brief commentary on this results). The methods say the data was gathered so the authors should provide that basic data.
Line 25 - the sentence starting on this line should read "The bacterial populations in the feces 5 days and 56 days after birth were not affected by the inclusion of feed additives."
Line 33 - insert "in" between mortality and newborn
Line 36 - replace "must" with "may"
Line 39 - A new paragraph should commence with the sentence beginning "Probiotics are ..."
The authors should use the definition of probiotics provided in the literature (eg. see Hill, C., Guarner, F., Reid, G. et al. The International Scientific Association for Probiotics and Prebiotics consensus statement on the scope and appropriate use of the term probiotic. Nat Rev Gastroenterol Hepatol 11, 506–514 (2014). https://doi.org/10.1038/nrgastro.2014.66)
Line 44 - use "According to Ohashi et al (2009) ... "
Line 56 - Bacillus should be in italics.
Line 59 - Bifidobacterium should be in italics
Lines 67 and 68 - the text should be altered as suggested above.
Line 80 - "A total of 90 Holstein heifers were studied from birth to 56 days of age."
Lines 83 to 84 - "... where they were housed until the age of 56 days."
Line 110 - Should read "The calves were provided with proper basic care throughout the study."
Line 110 - I'm not familiar with the term "render". I would use "carer".
Line 112 - I'm not familiar with Pederipra Sprey - should that be Spray?
Line 116 - the description of the cauterisation procedure (presumably de-budding) should be a new paragraph after line 122.
Line 147 - "... and were not treated with ..." (replace was with were)
Line 164 - remove the word "moreover"
Line 222 - remove "+ 6.18"
Line 223 - see the suggested change provided above.
Line 224 - should read "There was no difference between the groups in the incidence or duration of diarrhoea diseases." There is no need to quote the p level which presumably relates to the incidence of diarrhea (particularly as no data is provided except for the duration of the diarrhea which has a different p value). Why didn't the authors provide the incidence of diarrhoea data? I recommend it be added to Table 1.
Line 243 - Table 2. This table does not provide quantitate data. The title should be "Bacteria found in calf faces ...". The data is semi-quantitative at best.
Lines 248 to 257 - Is this one paragraph? The formatting needs to be addressed.
Line 248 - remove the word "species"
Line 250 - I suggest the sentence should be altered to " The amount of Campylobacter jejuni in the 3 groups increased across the course of the study."
Line 252 - after this line I suggest the authors comment on the disappearance of Morganella.
Line 256 - should read "There was no difference between the 3 groups in the bacterial populations found in the calf faces."
Line 281 - should read "... nitrogenous substances..."
Line 320 - please refer to the literature by citing the first author et al then the number in square brackets. Here there are 4 studies that should be cited as Knowles et al [34], etc.
Discussion and Conclusions - these sections need to be re-written. More careful attention to the writing of these sections is required. For example, in line 290 there is a sentence that is not a sentence ("Similar results reported [14]."). I found the discussion of diarrhoea disease in the study was in conflict with the conclusion made on diarrhoea observed in the study.
I recommend the authors consider the findings made and elaborate on these in the Discussion. Also, please examine the results more extensively and provide some commentary on the value of adding probiotics to the diet of growing calves. Careful attention must be paid to the interpretation of the results and the conclusions drawn from the results.
Author Response
Dear Reviewer,
thank you very much for the opponent's review of our study. We have tried to modify the study according to your instructions. All comments were substantive and improved the quality of this study. We hope that everything will be fine now.
Regards
Collective of Authors.

Reviewer 2 Report
General comments
If the topic is not really innovative, but the paper is written in good English, the presentation is serious, the bibliography , design, material and methods and discussion correct, even if some points deserve to be improved.
If the link between probiotics and pathogen in feces is obvious (table 2), my main concern regarding the study is that the interest in nitrogen metabolism (table 3) is not really supported by bibliographic data: why focus only on nitrogen and not on energy for example? What is the link between bifidobacteria, enterobacteria, lactic bacteria and nitrogen metabolism? Can you find bibliographical evidence justifying the measurements made? This point needs to be improved in this paper especially in the introduction and discussion since some results (urea) are significant or close to significance.
Second important concern : feed intake measurement is missing, and avoid clear conclusions related to nitrogen metabolism (see my comments lin 320). Any way to evaluate, at least milk replacer intake ?
Another point : taking into account the huge variability of results in calves, I consider that a tendency (P < 0,10) should be reported in the study (naturally with all the usage precautions : i.e. "nitrogen excretion in feces at 21st days of age tended to be significant").
Review in details
Line 22 the probability (P=0.0725) is not in agreement with the data in Table 1 (P=0.1957) can you clarify this point ? If the real value is P =0,0725, this is a tendency and the conclusions of your article must be nuanced.
Line 49 I suggest in "adult ruminants" instead of "in adults".
Line 56-59 since "Bacteria of the genus..." this reference to bacillus does not bring anything to the presentation of the subject because bacilli are not tested in the study. Would suggest to skip it.
Line 228 and 276 : the statistical analysis appears to be correct, but the data in Tables 1 and 3 would benefit from improvement. I suggest giving for each experimental group the average accompanied by letters showing a different meaning between groups (if P<0.05), adding a column for SEM (standard error to the mean), leaving column P and removing the "Significance" column, as such :
| Variables | n | BB | LEC | C | SEM | P |
Line 256 : "analysis" instead of "analyzis" to be in conformity with material and methods
Line 320 : the discussion must be improved in accordance with former remarks regarding the justification on your focus on nitrogen metabolism in line with probiotic usage. You mention "urea vary and may be related to protein intake in feed".... Did you measure the feed intake of calves. Or do you have a way to evaluate feed intake ? The better growth rate should be linked to better feed intake.
Line 330 : the P mentionned is not in line with the data of the table 1, same remarks as above (line 22).
Line 334 : "there was an improvement of health status" is contradictory with the lack of effect reported on line 330, need to be changed or reformulation. If you confirm that is a tendency (P<0,10) I would propose "health status tended to be improved"
Author Response
Dear Reviewer,
thank you very much for the opponent's review of our study. We have tried to modify the study according to your instructions. All comments were substantive and improved the quality of this study.
To your question: why focus only on nitrogen and not on energy for example? This is a very good reminder. Unfortunately, we did not monitor energy levels in this study, but we are currently preparing another trial focusing on prebiotics and a combination of probiotics and prebiotics. Here we will also monitor energy, because we will test different amounts of prebiotics (inulin, FOS and others), which are directly related to energy consumption and serve as an energy source for probiotics.
We hope that everything will be fine now.
Regards,
Collective of Authors.

Round 2
Reviewer 1 Report
Dear Authors
I am mostly happy with the revisions as shown in the revised manuscript.
I make the following suggestions:
Line 23 - change to "... unaffected except for the first sampling of urea."
Line 62 - change to "Ruminant animals ..." (note, not Ruminants)
Line 82 - change to "... 56 days of age, and used as experimental animals." Note, no full stop after the word 'age' and change from 'were" to "and".
Line 91 - lower case "t" in "the" after the first comma.
Line 141 - change to "... receiving a combination of ..."
Line 163 - change to "The carers monitored ..."
Line 268 - change to "... the intestine, can also be found in the surrounding environment."
Line 277 - replace the last word "faces" with "feces"
Line 288 - change LAB to LEB
Lines 310 and 311 - change to "... unaffected except for the first sampling of urea."
Line 320 - insert comma between "weight" and "daily"
Line 323 - remove the word "works"
Line 337 - insert "risk of" between "smaller" and "infection"
Line 354 - change "vary" to "varies"
Line 359 - insert the word "suggest:" between "that" and "different"
Line 364 - change LAB to LEB
Line 368 - I would put brackets around (Hammon et al. [46])
Line 371 - I would putt brackets around (Hammon et al. [49])
Line 384 - change to "... improved in experimental groups."
I make the above suggestions to improve the use of the English language in the manuscript. These suggestions are made to improve the manuscript's use of English and bring it to the standard expected of the journal, Antibiotics.
I appreciate the work done to bring the manuscript to this point. I strongly recommend you incorporate the minor changes I have suggested here.
Kind regards
Reviewer.
Author Response
Dear Reviewer,
thank you very much for your comments, we have included them all in the article. Changes to the article based on your comments are marked in green. Thank you for your time and feedback.
Kind regards
Collective of authors.
